# The Mindful Reappraisal of Pain Scale: Initial Validity Evidence for Use as a State Measure Following Isometric Exercise

**DOI:** 10.3390/bs15111586

**Published:** 2025-11-19

**Authors:** Sara A. Thompson, Reese Small, Anne E. Cox, Sarah Ullrich-French

**Affiliations:** 1Faculty of Kinesiology and Physical Education, University of Toronto, 55 Harbord St, Toronto, ON M5S 2W6, Canada; 2College of Education, Sport & Human Sciences, Washington State University, P.O. Box 1410, Pullman, WA 99164-1410, USA

**Keywords:** mindfulness, pain reappraisal, exercise, affect, pain tolerance

## Abstract

The Mindful Reappraisal of Pain Scale (MRPS) measures the capacity to reinterpret pain mindfully, supporting resilience and persistence during discomfort. The MRPS may be especially useful in acute, exercise-related pain contexts, where individuals experience short-term but physically demanding discomfort. This study is the first to evaluate the validity of a state-based version of the MRPS in an acute exercise context. Physically active participants (*N* = 127) completed a plank and wall sit and reported state mindful reappraisal of pain, trait and state mindfulness, and exercise experience perceptions. Confirmatory factor analyses supported a unidimensional structure with good internal consistency (ω = 0.88/0.90). MRPS scores correlated with mindfulness, pain tolerance, and affect and uniquely predicted wall sit pain tolerance after controlling for mindfulness. Scores were unrelated to pain intensity and perceived exertion, supporting the theoretical distinction of cognitive–affective reinterpretation rather than sensory attenuation. These findings support the MRPS as a brief, reliable tool for assessing mindful reappraisal in acute exercise contexts while also aligning with emerging evidence from clinical validation studies. However, further research is needed to confirm psychometric robustness across diverse exercise modes and participant populations.

## 1. Introduction

Pain is a barrier to exercise continuation ([26]). Exercise is inherently uncomfortable and induces physiological stress, leading to interoceptive signals such as increased arousal, elevated heart rate, and fatigue ([10]). As exercise intensity increases, these interoceptive signals become increasingly perceived as unpleasant, particularly when surpassing the ventilatory threshold ([12]; [21]). Such unpleasant experiences can create negative associations with exercise, contributing to dropout, avoidance, or discontinuation ([4]). Therefore, identifying strategies that help individuals positively reframe or manage these sensations is crucial for promoting long-term physical activity engagement.

The perception of pain is not purely a sensory event but is heavily shaped by cognitive appraisals and consequent emotional reactivity ([38]; [37]). Negative appraisals of pain tend to amplify suffering and lead to maladaptive responses that reinforce avoidance behaviors (e.g., fear of movement or catastrophizing) ([34]). These cognitive–emotional loops contribute to the persistence of pain and the diminished efficacy of pharmacological interventions, which often fail to target the neural circuits underlying pain appraisal and negative affective amplification, as evidenced by the limited efficacy of common analgesic drugs for numbing pain ([7]). Consequently, exploring the pain appraisal process is a promising pathway for effectively managing pain, particularly during common painful experiences such as exercise.

Pain can be managed in different ways. Distraction is a commonly employed cognitive strategy for pain management, involving the intentional diversion of attention away from painful stimuli toward different tasks or thoughts. This approach can temporarily reduce perceived pain by occupying cognitive resources that would otherwise process nociceptive signals ([27]). However, the effectiveness of distraction is inconsistent and context-dependent. In situations where attention to bodily sensations is necessary, such as during exercise, distraction may not be feasible or beneficial ([24]). Moreover, reliance on distraction can impede executive functioning, which may increase the perception of pain, potentially leading to long-term challenges in pain management ([33]). Therefore, while distraction can be a useful tool in certain situations, it is important to consider its limitations and the potential benefits of alternative strategies that promote active engagement with pain experiences.

Mindfulness, defined as paying attention to the present moment with openness, curiosity, and acceptance, has emerged as a promising nonpharmacological approach to pain management ([22]; [2]). Mindfulness-based interventions (MBIs) focus on enhancing bodily awareness and fostering an accepting attitude toward various thoughts, emotions, and sensations without judgement. Research has demonstrated that MBIs can have analgesic effects and reduce pain perception (see [15]; [19]). However, the mechanisms by which MBIs exert their effects on pain perception are not fully understood ([16]). Most existing mindfulness interventions emphasize acceptance-based approaches, but there is limited evidence to suggest that the mere acceptance of pain effectively reduces its severity ([11]).

Recently, [16] ([16]) have taken a new approach to managing pain perceptions by exploring mindful pain appraisals that shift focus from acceptance-based mindfulness to the reinterpretation of pain experiences within a mindful framework. This concept integrates attentional awareness and distancing. Together, these two attentional stances toward pain allow for a more adaptive cognitive interpretation of pain. This is a key component of the Mindfulness-to-Meaning Theory (MMT; [14]), which suggests that mindfulness practice can catalyze a positive reappraisal process through shifts in attention and awareness. According to MMT, mindfulness enables individuals to step back from automatic negative interpretations of pain or stress and instead reflect on these experiences in a more open and constructive way. This shift can lead to assigning new meanings to difficult sensations, such as viewing pain as temporary, a sign of strength, or a natural, non-harmful experience. Over time, these reappraisals can support emotional regulation, promote positive emotions, and increase psychological resilience ([14]).

To test the key mechanisms of mindfulness according to MMT, [16] ([16]) adapted the Reinterpreting Pain Sensations subscale from the Coping Strategies Questionnaire ([28]) to better align the item language with MBIs. Items not aligned with mindfulness were modified and new items were generated to better capture mindfulness. For example, the item “I pretend it’s not a part of me” was modified to “I watch my pain from a distance, as if I were an objective observer.” The result was a new nine-item scale called the Mindful Reappraisal of Pain Scale (MRPS: [16]). Through two independent samples of individuals with chronic pain, including adults with chronic musculoskeletal pain (*n* = 450; *n* = 95), evidence supports a unidimensional factor structure for the nine-item MRPS. Internal consistency reliability was good (alpha = 0.84, Omega = 0.84), showing initial evidence for the psychometric properties of the MRPS. Correlations with relevant variables further supported convergent and discriminant validity. The MRPS was positively correlated with reinterpretation of pain sensations and cognitive reappraisal, interoceptive awareness (self-regulation and body listening), and the observing facet of trait mindfulness. There were weak correlations with other facets of trait mindfulness, suggesting the MRPS is a related but distinct construct from trait mindfulness.

The MRPS is a promising tool for measuring the mechanisms underpinning the effects of mindfulness on pain. However, the scale has primarily been used to assess trait levels in adults with chronic pain. Chronic pain and acute pain differ in duration, appraisal demands, cognitive load, and underlying mechanisms, and therefore a scale validated in chronic contexts may not generalize to short, performance-related pain states. Therefore, there is a need to validate the MRPS in a more dynamic, state context, such as acute pain experienced during exercise. Validating the MRPS in an exercise setting is necessary to determine whether mindful reappraisal can be reliably measured during brief, high-arousal pain states, where attentional demands, physiology, and motivational context differ from chronic pain. By measuring state mindful reappraisal of pain in response to physical exertion, we may better understand how individuals perceive the discomfort of exercise and the mechanisms underlying the efficacy of MBIs in exercise settings. Isometric exercises, which involve muscle contractions without joint movement, are used in experimental pain research because it can reliably induce muscular discomfort or fatigue within a short time frame and can be standardized across individuals by controlling for contraction intensity and duration ([32]). Studying mindful reappraisal of pain during isometric exercise could be valuable, but it remains unclear how the MRPS performs in an exercise setting or as a state measure.

The purpose of this study was to evaluate validity evidence for the MRPS in a state format during isometric exercise. We tested whether the MRPS items demonstrated a one-factor structure consistent with [16]’s ([16]) confirmatory factor analysis (CFA) and evaluated construct validity by testing associations between MRPS scores and conceptually related variables, including trait mindfulness, state mindfulness during exercise, affect, and pain tolerance. Lastly, we explored whether MRPS scores explained variance beyond trait and state mindfulness scores, indicating that mindful reappraisal reflects a distinct cognitive–affective process during physical discomfort. Together, these aims provide initial evidence for the use of the MRPS in acute, embodied pain contexts and lay the groundwork for future research on mindful pain coping in physically active populations.

## 2. Method

### 2.1. Procedures

Baseline data from two samples using the same baseline procedures were used for this study. In both studies, participants completed a baseline assessment of trait mindfulness prior to completing two isometric exercises (plank and wall sit). Exercise order was randomly assigned to reduce systematic differences across participants. All participants completed a Physical Activity Readiness Questionnaire (PAR-Q) to screen for any injuries or health conditions that would limit safe participation in exercise. Participants were all considered physically active by either self-report or by current participation in a 16-week physical activity class. All participants were instructed on the proper form for a plank and a wall sit and given a chance to practice proper form for 5 s as a warmup. Participants were then instructed to hold each exercise as long as possible while maintaining form, as directed by a partner or an instructor. After the first exercise, participants completed state assessments about the exercise experience, including state mindful reappraisal of pain, state mindfulness, RPE, pain during exercise, and core affect during exercise. After a four-minute break, participants completed the second exercise followed by state assessments about the second exercise experience. To minimize fatigue, only one trial of each exercise was performed, and participants received a standardized rest (must wait a minimum of 4 min and have heart rate be around 50% of the individual max heart rate) between exercises. Motivational decline was not directly assessed; however, both exercises involved comparable cognitive demands, and exercise order was counterbalanced to minimize systematic order effects. The research procedures for study one was approved by the Washington State University IRB (#200112-001, approved 15 September 2023), and study two’s procedures were certified as exempt by the Washington State University IRB (#20446-001, 11 March 2024).

### 2.2. Participants

Participants from study one (*N* = 55) and study two (*N* = 72) were combined for a total sample of 127 participants. The studies were conducted at the same university, used the same baseline procedures, and recruited from the same population. As the eligibility criteria for both studies required participants to be currently physically active, all participants met physical activity guidelines of at least 150 min of moderate-intensity exercise or 75 min of vigorous-intensity exercise per week. Race/ethnicity for study one was reported as White (61.8%), Asian/Pacific Islander (18.2%), Multiracial (10.9%), Hispanic/Latino (7.3%), and Other (1.8%). For study two, participants self-identified as White (44.6%), Asian/Pacific Islander (3.1%), Multiracial (12.3%), and Hispanic/Latino (9.2%), with 30.7% not reporting race. The two samples did not differ on age (t = 0.10, *p* = 0.92), gender (χ_2_ = 2.97, *p* = 0.40), or trait mindfulness (t = 1.26, *p* = 0.21), indicating that the samples were highly similar. The combined sample ranged in age from 18–43 (*M* = 22.01, *SD* = 4.47) and self-identified as female (48.8%), male (47.2%), other (1.6%), or preferred not to respond (0.8%). Responses on the MRPS were not recorded for one participant during the plank and one different participant for the wall sit. Therefore, the factor structure was tested on 126 participants for the plank and wall sit, respectively. The sample size was considered acceptable because the MRPS has 9 items and is expected to fit a one-factor model (a relatively simple confirmatory factor analysis model) that has been supported by sample sizes similar to ours, which is a good indicator of adequate sample size ([36]), and meets the general rule of greater than 10 cases per indicator (*N* = 126 > 10 * 9 items) ([35]). Power analysis using G*Power 3.1.9.7 suggests a sample size of 82 for two-tailed correlation tests with a medium effect size (0.30), α = 0.05, and power = 0.80.

### 2.3. Measures

Trait Mindfulness: Participants’ daily mindfulness habits were assessed using the short-form Five-Facet Mindfulness Questionnaire (FFMQ-SF; [1]), which includes 15 questions on a 5-point scale ranging from 1 (*very rarely true*) to 5 (*very often*). Responses were averaged to yield a single trait mindfulness score, and subscale scores were calculated for observing, describing, acting with awareness, non-judgment, and non-reactivity. The FFMQ-SF has been used in adult samples and has demonstrated good evidence of reliability and validity ([31]).

State Mindfulness: Participants completed the State Mindfulness Scale for Physical Activity 2 (SMSPA-2; [30]) immediately following the completion of each exercise. The SMSPA-2 uses a 5-point rating scale to measure participants’ monitoring/awareness and acceptance/nonjudgement of their physical and mental experience during each exercise. The scale ranged from 1 (*not at all*) to 5 (*very much*). All items averaged for a total score reflecting state mindfulness, and subscale scores were calculated for mental awareness, body awareness, mental acceptance, and body acceptance. This scale has demonstrated evidence supporting reliability and validity in adult samples within acute exercise contexts ([30]).

Exercise Pain Tolerance: Pain tolerance was defined as the maximum duration (in seconds) participants could hold each isometric exercise (forearm plank and wall sit). Time was recorded from the start of the exercise until the participant voluntarily stopped or was asked to stop due to improper form. Second-session maximum hold times were compared to baseline to evaluate changes in tolerance. Maximum hold time is a validated proxy for pain tolerance and muscular endurance in exercise contexts ([13]).

Affective Response to Exercise: Affective response to each exercise was assessed by participants rating their feelings of pleasure or displeasure using the 10 point Feeling Scale, ranging from −5 (*very bad*) to +5 (*very good*) (FS; [17]). Participants in sample one reported affect during each exercise using the Feeling Scale, and then the ratings were averaged. Participants in sample two reported one Feeling Scale rating of affect immediately after the completion of each exercise and were instructed to give their rating for the hardest part of the exercise. The Feeling Scale is validated to assess affect during physical activity ([29]).

Exercise Pain Intensity: Pain intensity experienced during each exercise was assessed using the 11-point Numeric Rating Scale (NRS; [39]) during exercise (sample one) and immediately post-exercise (sample two) (0 = no pain; 10 = worst pain possible). The NRS has valid evidence for use in isometric exercise contexts ([18]).

I The Borg Ratings of Perceived Exertion Scale (RPE; [3]) was used to assess perceived effort immediately post-exercise, focusing on the peak intensity of effort (6 = *no exertion at all* and 20 = *maximal exertion*). The RPE scale is widely used with validity support for adult populations ([3]).

### 2.4. Data Analysis

Factor Structure. A confirmatory factor analysis (CFA) using a robust maximum likelihood estimator was run on the MRPS items for plank and wall sit, respectively, using MPlus 8. We tested the hypothesized one-factor model, where all items loaded on one factor because the original MRPS factor structure and theoretical rationale fit a one-factor structure ([16]). Model fit was assessed using the root mean square error of approximation (RMSEA), standardized root mean square of residual (SRMR), Tucker–Lewis fit index (TLI), and comparative fit index (CFI) using the [20] ([20]) fit criteria. Aligned with [16]’s ([16]) analysis, we allowed the errors to correlate among similar items.

Construct Validity Tests Using MRPS-Calculated Scores. Scores were calculated based on the CFA results for the MRPS items. Descriptive statistics, omega, and alpha internal consistency reliability were calculated for the MRPS and other multi-item scales. Missing data was minimal (<4%), and Little’s MCAR test indicates the data were missing at random (χ^2^ = 88.56, DF = 81, *p* = 0.27). Therefore, we used expectation maximization to impute missing data. To assess construct validity, Pearson’s correlations were calculated among all scores to assess how the MRPS scores associated with conceptually relevant variables. We explored evidence for discriminant validity and unique variance by running partial correlations after controlling for trait and state mindfulness. Analyses were conducted for plank and wall sit experiences separately.

## 3. Results

### 3.1. Factor Structure

We tested one-factor CFA models for state MRPS during plank and wall sit, respectively. CFA results supported the one-factor structure. Model fit for plank MRPS had good fit (CFI = 0.95, TLI = 0.92, RMSEA = 0.08, SRMR = 0.05). Factor loadings (0.49–0.84) were significant (*p* < 0.001). Model fit for wall sit MRPS showed good fit (CFI = 0.95, TLI = 0.92, RMSEA = 0.08, SRMR = 0.04). Factor loadings (0.50–0.80) were significant (*p* < 0.001). See Table 1 for CFA item loadings.

### 3.2. Calculated MRPS Scores and Correlations with Exercise Experience

The average MRPS score was calculated for the plank and wall sit exercise experience, respectively. Table 2 shows means and standard deviations for all variables. All variables were approximately normal in distribution (skewness and kurtosis all <0.5). The internal consistency reliability was good for plank (ω = 0.88/α = 0.88) and wall sit (ω = 0.90/α = 0.90) MRPS (See Table 3 for reliability of all scales). See Table 4 for bivariate correlations. State MRPS was positively correlated with trait mindfulness, state mindfulness, pain tolerance, and affect during exercise. MRPS was not related to pain during exercise or RPE for wall sit. The exercise experience associations follow expected patterns, where RPE was positively associated with pain during exercise and negatively associated with affect during exercise. Trait mindfulness was moderately positively associated with MRPS, state mindfulness, and affect during exercise. The MRPS construct was the only mindfulness-related variable significantly associated with pain tolerance for the wall sit, and both MRPS and state mindfulness associated with pain tolerance for the plank.

The correlations among the trait and state mindfulness subscale scores were explored to identify how different facets of mindfulness relate to MRPS scores for comparison to [16] ([16]) (see Table 4(a,b)). The MRPS scores were associated with the observing and non-reactivity facets of trait mindfulness for both the plank and the wall sit. The MRPS scores were associated with all subscales of state mindfulness, with the exception of plank body awareness. The correlations were of small-to-moderate magnitude.

Partial correlations were calculated for all mindfulness constructs to control for trait and state mindfulness as well as state mindful reappraisal of pain to assess the unique variance in the MRPS correlations after controlling for other mindfulness constructs (see Table 5). MRPS association with plank tolerance did not result in significant unique variance. The association of MRPS remained significantly unique with plank affect after controlling for state mindfulness (*R*^2^ = 0.04). There was strong MRPS unique variance (*R*^2^ = 0.13) with wall sit tolerance that was not reduced controlling for trait and state mindfulness and was marginally reduced for wall sit affect (*R*^2^ = 0.04), showing meaningful unique variance in both constructs. MRPS associations with wall sit pain (*R*^2^ = 0.03) and RPE (*R*^2^ = 0.03) showed moderate unique variance. Unique variance of trait mindfulness with affect during the plank (*R*^2^ = 0.13) and wall sit (*R*^2^ = 0.04) were not meaningfully reduced controlling for state mindfulness and/or MRPS. There was no significant or meaningful unique variance for state mindfulness.

## 4. Discussion

This study sought to evaluate the factor structure and construct validity of the Mindful Reappraisal of Pain Scale (MRPS) in an acute exercise setting. Consistent with [16] ([16]), the results supported a unidimensional factor structure for the MRPS, replicating strong internal consistency in this novel, state-based task of a wall sit and forearm plank. The results of the CFA support the use of a single score with good internal consistency and reliability. Psychometric results were highly consistent with [16] ([16]). Correlations further support the construct validity of the scale used to assess acute MRPS in exercise. Partial correlations further support meaningful unique variance explained with MRPS scores. As initial validity evidence is a subset of a measure’s psychometric properties, these findings indicate preliminary psychometric support for the MRPS in an acute exercise context, suggesting that the scale, which was originally validated in chronic pain populations, may also be a reliable and valid measure of mindful reappraisal during acute, exercise-related pain states. This application is theoretically supported by the Mindfulness-to-Meaning Theory, which posits that reappraisal unfolds dynamically in response to changing bodily sensations ([14]). Thus, even though the MRPS was developed in chronic pain populations, the core mechanism is the reinterpretation of sensory discomfort in adaptive ways, which extends conceptually to acute exertional pain.

The pattern of correlations shows alignment on constructs that are conceptually similar as expected. Trait mindfulness is moderately correlated with state mindfulness assessments, aligning with past research that shows a similar moderate correlation between state and trait mindfulness ([30]). It is expected to have some overlap in mindfulness from trait to state, but there is also room for distinct experiences that are situation specific (see [9]). The results show this with trait mindfulness being moderately correlated with both state mindfulness and the state assessed MRPS. Similar to [16] ([16]), correlations with the mindfulness facets were most pronounced for the observing facet and the non-reactivity facet but unrelated to the other specific facets. Although the marginal reliability of the observing subscale should be noted. Trait mindfulness did not correlate with most of the experiential assessments (e.g., RPE, pain tolerance) but did correlate positively with affect during exercise for the total score, all facets for the plank, and just the observing and non-reactivity facets for the wall sit. Having higher trait mindfulness associated with higher state mindfulness, higher mindful reappraisal of pain, and higher positive affect during exercise. The state assessed MRPS did associate with pain tolerance, suggesting that the MRPS captures a unique and context-specific cognitive–affective process that unfolds during pain-induced activities, which aligns with the Mindfulness-to-Meaning Theory ([14]).

MRPS scores predicted pain tolerance during the wall sit exercise even when controlling both trait and state mindfulness. This finding highlights the potential functional role of mindful reappraisal of pain in modulating persistence in the face of discomfort, over and above general trait mindfulness. Whereas traditional mindfulness metrics capture general tendencies or present-centered awareness, the MRPS appears to have a mechanism that actively reframes discomfort and enables continued engagement with effortful tasks, such as isometric exercise. This ability to maintain presence with pain while cognitively restructuring its meaning (e.g., pain as a sign of strength or progress) may be especially critical in exercise contexts where discomfort is often misinterpreted as a threat or signal to disengage.

Interestingly, MRPS scores were not systematically associated with pain intensity or RPE. This aligns with the Mindfulness-to-Meaning Theory, which suggests that reappraisal does not alter the sensory dimension of pain but instead modifies its emotional meaning and impact ([14]). The affective findings further support the MRPS’s utility. Higher MRPS scores were associated with more positive affect during exercise, and these associations held for the wall sit after controlling for trait and state mindfulness. This suggests that mindful reappraisal is not only linked to better persistence but also to more adaptive emotional outcomes, potentially ‘buffering’ the negative affective consequences of physical discomfort. This is consistent with prior findings that mindfulness fosters emotion regulation by reducing maladaptive cognitive–affective loops ([14]). This is also consistent with muscular endurance exercise performance research, suggesting that physically trained individuals may be able to use sensory feedback more adaptively by being able to regulate effort while experiencing physical discomfort ([6]) and reducing negative emotional responses during exercise ([5]). Understanding these processes are important because sensory feedback of pain is a determinant of effort maintenance and muscular fatigue and possible discontinuation ([25]; [23]). Ultimately, interventions teaching mindful pain reappraisal could be fruitful for exercise performance and persistence over time.

The exercise experience assessed through RPE, pain, and affect show correlations expected by the Affective Reflective Theory (ART) of exercise ([4]). According to ART, as exercise intensity increases, so do the interoceptive bodily cues (e.g., physical discomfort/pain). The affective experience with higher levels of intensity and discomfort are less positive. We saw this pattern in the participant’s exercise experiences, where higher levels of RPE related to higher pain and lower positive affect. Moreover, higher pain was associated with lower positive affect. Interestingly, the MPR scores did not systematically vary with pain or perceived exertion but did associate with the affective response to acute isometric exercise. The conceptual process detailed by [16] ([16]) in developing the MRPS is consistent with these findings. Mindfulness brings awareness, which is why general pain experienced and level of perceived exertion are not related to MPR. However, the ability to hold that awareness with a more adaptive interpretation could explain why those with higher MPR scores also reported more positive affect and higher pain tolerance. These results are preliminary, and further research is needed to test the use of the MRPS with larger and more diverse samples and in different exercise modes.

A strength of this study is its ecological relevance. While prior MRPS validation studies were conducted in clinical contexts, the current study investigated mindful reappraisal within physically demanding tasks that are typical isometric exercises. This has meaningful implications for exercise psychology and behavioral medicine. Given that negative experiences during exercise are a key barrier to long-term adherence ([12]), interventions targeting mindful reappraisal could help individuals tolerate discomfort more effectively and sustain engagement in physical activity routines. The MRPS, particularly in its state-based adaptation, may serve as a valuable tool for evaluating the effectiveness of such interventions or for identifying individuals who might benefit from cognitive–affective training. Practically, the MRPS could be used to monitor changes during mindfulness or reappraisal-based interventions or to inform adaptive exercise training programs that tailor difficulty and coaching strategies based on an individual’s cognitive-affective response to discomfort. In exercise settings, MRP scores may help identify participants who are more vulnerable to negative affective responses or early task discontinuation, allowing instructors to provide targeted coaching or reappraisal prompts. The measure may also inform adaptive training programs by adjusting exercise intensity or psychological support in response to an individual’s cognitive–affective profile. As such, the MRPS offers a brief, scalable tool for evaluating and optimizing both psychological and physical training processes.

### Limitations

There are limitations of this study. First, the sample size was limited, and replicating the CFA results for the MRPS with larger samples will provide a stronger basis of evidence for the scale. The sample was relatively young, healthy, and physically active, which limits the generalizability of the results. Further research exploring the MRPS with more diverse samples is important. Additionally, although the isometric exercise model offers reliable and controlled pain induction, future studies should assess whether the validity of the MRPS extends to other forms of physical activity.

Moreover, while self-report remains a practical method for assessing cognitive–affective processes, the self-report nature of the scale introduces limitations. Notably, self-report is vulnerable to response biases, particularly in the context of mindfulness, where social desirability or self-awareness may be heightened. However, the ability to have a short scale that is reliable and can explain variance in exercise experiences serves as a valuable tool for future research. Another limitation is that the study did not include physiological markers of exertion or metabolic stress (e.g., blood lactate or oxygen consumption), which could complement self-reported pain and perceived exertion and provide a more comprehensive profile of the exercise response. Future work could incorporate physiological or behavioral correlates of mindful reappraisal, such as heart rate variability, EMG, blood lactate accumulation, or task persistence under deceptive or ambiguous feedback conditions.

This study design did not manipulate reappraisal directly. While individual differences in MRPS scores were meaningfully associated with outcomes, experimental designs (e.g., reappraisal instruction vs control) are needed to establish causal relationships. Indeed, such designs would help determine whether boosting reappraisal abilities via brief interventions can enhance pain tolerance and emotional resilience in acute settings. In addition, it would be valuable to examine whether MRPS scores mediate the relationship between mindfulness interventions and behavioral outcomes, such as exercise adherence, motivation, or performance over time. This would clarify whether mindful reappraisal serves as a pathway linking mindfulness practice to improved physical and psychological functioning. Future work should also distinguish mindful reappraisal from other cognitive coping strategies (e.g., distraction or suppression strategies) to determine whether reinterpretation of discomfort offers advantages over other alternative approaches. Finally, in CFA models with a relatively small sample, the standard errors could be biased, and generally, the quality of goodness-of-fit tests may be questionable. However, parameter estimates are less likely to be biased because we did not face non-convergence and improper solutions problems during model estimation ([8]). Although preliminary, the results are supportive of further tests of the MRPS in acute exercise contexts.

## 5. Conclusions

In summary, the present study provides initial evidence supporting the use of the Mindful Reappraisal of Pain Scale as a reliable and valid tool for assessing state-level cognitive reappraisal of pain in acute exercise contexts. The scale’s factor structure and construct validity replicated those found in chronic pain populations, and its associations with affect and pain tolerance suggest it captures a distinct, functional process relevant to how individuals interpret and respond to physical discomfort. By shifting the application of the MRPS from static, trait-based assessments to dynamic, embodied experiences, this study extends the scale’s utility and opens new avenues for understanding how mindful reappraisal may facilitate resilience and persistence in the face of aversive bodily sensations. The MRPS holds promise for both research and practice as a concise, theoretically grounded measure of adaptive pain coping in physically active populations. However, future research should examine whether MRP scores predict behavioral outcomes such as exercise adherence, persistence, or performance across longitudinal or intervention-based studies.

## Figures and Tables

**Table 1 behavsci-15-01586-t001:** Confirmatory factor analysis factor loadings for plank and wall sit MRPS items.

	Plank			Wall Sit		
MRPS Items	Factor Loading	SE	*R* ^2^	Factor Loading	SE	*R* ^2^
1	0.71	0.06	0.55	0.80	0.04	0.64
2	0.77	0.06	0.59	0.79	0.05	0.63
3	0.68	0.07	0.36	0.50	0.09	0.25
4	0.64	0.07	0.45	0.69	0.07	0.48
5	0.62	0.08	0.44	0.68	0.07	0.46
6	0.49	0.09	0.25	0.61	0.07	0.38
7	0.84	0.05	0.61	0.76	0.07	0.57
8	0.54	0.10	0.28	0.70	0.06	0.49
9	0.67	0.07	0.49	0.71	0.05	0.50

*Note.* All factor loadings were significant (*p* < 0.001). MRPS = Mindful Reappraisal of Pain Scale; SE = standard error; *R*^2^ = coefficient of determination.

**Table 2 behavsci-15-01586-t002:** Descriptive statistics.

		Plank		Wall Sit	
	Scale	Mean	SD	Mean	SD
MRPS	1–7	3.13	1.21	3.17	1.25
State Mindfulness	0–4	2.48	0.51	2.48	0.58
Pain Tolerance	(seconds)	102.17	42.15	95.51	48.88
RPE	6–20	14.05	1.93	14.33	2.08
Exercise Pain	0–10	4.10	1.91	4.77	2.11
Exercise Affect	−5 to 5	0.43	2.16	0.32	2.33
		**Mean**		**SD**	
Trait Mindfulness	1–5	3.27		0.48	

**Table 3 behavsci-15-01586-t003:** Scale reliability.

Scale	α	ω
Trait Mindfulness Total	0.75	0.72
Observing	0.63	0.64
Describing	0.83	0.83
Acting with Awareness	0.68	0.73
Non-Judgement	0.76	0.77
Non-Reactivity	0.66	0.76
Plank MRPS	0.88	0.88
Plank State Mindfulness Total	0.82	0.78
Mind Awareness	0.86	0.86
Body Awareness	0.79	0.79
Mind Acceptance	0.67	0.67
Body Acceptance	0.82	0.82
Wall Sit MRPS	0.90	0.90
Wall Sit State Mindfulness	0.83	0.85
Mind Awareness	0.89	0.89
Body Awareness	0.81	0.80
Mind Acceptance	0.61	0.66
Body Acceptance	0.88	0.86

**Table 4 behavsci-15-01586-t004:** Bivariate correlations. (**a**) Bivariate Correlations of Study Variables with MRPS; (**b**) Bivariate Correlations with Trait and State Mindfulness Subscales.

(**a**)									
			**1**	**2**	**3**	**4**	**5**	**6**	**7**
1 MRPS			0.86 **	0.33 **	0.43 **	0.18 *	−0.08	−0.08	0.25 **
2 Trait Mindfulness			0.37 **	−	0.27 **	0.16	−0.00	−0.14	0.42 **
3 State Mindfulness			0.51 **	0.27 **	0.76 **	0.19 *	−0.03	0.03	0.19 *
4 Tolerance			0.25 **	0.04	0.05	0.48 **	0.17 ^	0.07	0.06
5 RPE			−0.17	−0.08	0.01	0.06	0.66 **	0.46 **	−0.26 **
6 Exercise Pain			−0.12	−0.14	0.07	−0.10	0.50 **	0.71 **	−0.43 **
7 Exercise Affect			0.33 **	0.29 **	0.23 *	0.15	−0.25 **	−0.22 *	0.68 **
(**b**)									
	**Trait Mindfulness**					**State Mindfulness**			
	**OB**	**DS**	**AA**	**NJ**	**NR**	**MA**	**BA**	**MAc**	**BAc**
Plank									
MRPS	0.36 **	0.13	0.03	0.02	0.26 **	0.33 *	0.13	0.35 **	0.28 **
Trait	0.59 **	0.68 **	0.49 **	0.61 **	0.50 **	0.19 *	0.20 *	0.44 **	0.35 **
State	0.42 **	0.20 *	−0.05	−0.07	0.23 *	0.59 **	0.42 **	0.45 **	0.51 **
Wall Sit									
MRPS	0.43 **	0.16	−0.01	0.05	0.28 **	0.43 **	0.18 *	0.46 **	0.43 **
Trait	0.59 **	0.68 **	0.49 **	0.61 **	0.50 **	0.23 **	0.22 **	0.33 **	0.41 **
State	0.34 **	0.17 ^	0.01	0.01	0.20 *	0.56 **	0.47 *	0.50 **	0.61 **

*Notes*. * *p* < 0.05, ** *p* < 0.01, ^ *p* = 0.05. (**a**). Correlations on the upper diagonal are for plank and lower diagonal (shaded) are for wall sit. RPE = Rating of Perceived Exertion. (**b**). Trait mindfulness subscales include OB = observing, DS = describing, AA = acting with awareness, NJ = non-judgment, NR = non-reactivity. State mindfulness subscales include MA = mind awareness, BA = body awareness, MAc = mind acceptance, BAc = body acceptance.

**Table 5 behavsci-15-01586-t005:** Zero-order and partial correlations for MRPS, trait mindfulness, and state mindfulness.

	Tolerance		Pain During		Affect During		RPE	
Plank	Zero	Part	Zero	Part	Zero	Part	Zero	Part
MRPS (TM)	0.18 *	0.14	−0.08	−0.04	0.25 **	0.13	−0.08	−0.09
MRPS (SM)		0.11		−0.10		0.19 *		−0.08
MRPS (TM/SM)		0.08		−0.07		0.11		−0.08
Trait (MRP)	0.16	0.11	−0.14	−0.15	0.42 **	0.39 **	−0.01	0.01
Trait (SM)		0.11		−0.15		0.39 **		0.01
Trait (MRP/SM)		0.09		−0.13		0.36 **		0.03
State (MRP)	0.19 *	0.13	0.03	0.07	0.19 *	0.09	−0.03	0.00
State (TM)		0.16		0.07		0.09		−0.04
State (MRP/TM)		0.12		0.09		0.04		−0.00
Wall Sit								
MRPS (TM)	0.25 **	0.25 **	−0.12	−0.07	0.33 **	0.25 **	−0.17	−0.15
MRPS (SM)		0.26 **		−0.18 *		0.26 **		−0.20 *
MRPS (TM/SM)		0.26 **		−0.14		0.20 *		−0.18 *
Trait (MRP)	0.04	−0.05	−0.14	−0.10	0.29 **	0.19 *	−0.08	−0.02
Trait (SM)		0.03		−0.16		0.24 **		−0.09
Trait (MRP/SM)		−0.05		−0.12		0.19 *		−0.03
State (MRP)	0.05	−0.09	0.07	0.16	0.23 *	0.07	0.01	0.11
State (TM)		0.04		0.12		0.16		0.03
State (MRP/TM)		−0.09		0.17		0.05		0.11
(control)								

*Notes*. * *p* < 0.05, ** *p* < 0.01. “(TM)” indicates partial correlations controlling for trait mindfulness; “(SM)” indicates partial correlations controlling for state mindfulness; and “(TM/SM)” indicates partial correlations controlling for both trait and state mindfulness. “(MRP)” indicates partial correlations controlling the mindful reappraisal of pain score. MRP = mindful reappraisal of pain; TM = trait mindfulness; SM = state mindfulness.

## Data Availability

Data is available upon request to the corresponding author.

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
