# Peer review of "The Mindful Reappraisal of Pain Scale: Initial Validity Evidence for Use as a State Measure Following Isometric Exercise"

_behavsci, 2025, doi:10.3390/bs15111586_

Round 1

Reviewer 1 Report

Comments and Suggestions for Authors

Overall, this is an appropriate and interesting topic for the journal. The study addresses a highly relevant and timely issue, employing a methodological approach and analysis that provide important insights into the phenomenon under investigation. While the strengths of the study are evident, I have identified several areas that would benefit from further development and clarification. My specific comments, intended to improve the manuscript, are detailed below.

 At the end of the abstract, consider addressing the following question: Does this finding align with existing evidence in the literature, or are further studies needed to confirm the psychometric robustness of the MRPS in similar contexts?

Introduction: It is suggested that some of the references cited before 2015 be reviewed, as the phenomena discussed may be better supported or complemented by more recent reviews, thereby reinforcing the relevance of the theoretical framework.

Methods and Materials: Regarding the procedures and participants, did the authors verify  any differences between the two samples beyond the shared baseline procedures (e.g., demographics, fitness level, or context)? Was there any control for potential fatigue or motivational decline between the two exercises?
Specifically concerning participants, the authors should clarify whether they screened for prior injuries or health conditions that could influence pain tolerance or exercise performance.
Regarding the instruments, please clarify whether the reliability and validity of the state measures have been established in this context.
The authors state that the study procedures were approved by the Institutional Review Board; however, it would be helpful to include the approval number, reference, or the name of the IRB that granted approval.
What was the rationale for testing only a one-factor model?

Discussion:  The discussion section is well developed, but a few clarifications could further enhance its quality. For example, the MRPS was originally developed for chronic pain populations. In this context, please clarify which theoretical justification supports its direct adaptation to acute, performance-based discomfort. Additionally, it might be interesting for future work further differentiate between mindful reappraisal and other forms of cognitive coping. 

Final comment: A formal review of the reference list should be conducted to ensure strict compliance with the journal’s editorial standards

Reviewer 2 Report

Comments and Suggestions for Authors

This manuscript provides a valuable contribution to the field by evaluating the MRPS) in a novel, state-based context of acute exercise-induced pain. The methodological approach using isometric exercises is sound, and the analyses are generally appropriate and thorough. The findings offer promising initial evidence for the reliability and validity of the state MRPS in this setting. The manuscript is well-structured, but several points require clarification and refinement to strengthen the presentation and interpretation of the results.

Please address the following points to improve the manuscript:

  1. Abstract

Consider stating more explicitly that this is the first study to validate the *state* version of the MRPS in an acute exercise context.

  1. Introduction (P2)

The final paragraph would benefit from a more explicit and concise listing of the study's specific aims and hypotheses (e.g., regarding factor structure, correlations, and unique variance).

  1. Methods

Participants (P4): Justify the combination of the two samples by providing a brief comparison of their key demographic characteristics or baseline measures to demonstrate comparability.

Measures (P4-5): Report the internal consistency reliability coefficients (e.g., Cronbach's α or McDonald's ω) for all multi-item scales (e.g., FFMQ-SF, SMSPA-2) obtained in the current study.

Data Analysis (P5): Clearly state the pre-determined criteria used for judging acceptable model fit in the CFA (e.g., CFI/TLI > .90, RMSEA < .08, SRMR < .08).

  1. Results

Factor Structure / Table 1 (P5): The caption for Table 1 should explicitly note that all factor loadings were statistically significant (p < .001).

Calculated MRPS Scores / Tables 3, 4, 5 (P 6-7): The abbreviations used in these tables (e.g., TM, SM, MRP) need to be clearly defined in the table notes or a footnote to aid reader comprehension.

Calculated MRPS Scores / Table 5 (P7): The text or table notes should unambiguously specify which variables were controlled for in each partial correlation analysis (e.g., ‘MRPS (TM) signifies the correlation controlling for Trait Mindfulness).

  1. Discussion

P8, first paragraphs:  The explanation that MRPS scores were unrelated to pain intensity and RPE is somewhat repetitive across the first two paragraphs. This point could be consolidated for a more concise and impactful presentation.

P9, middle:Consider expanding the discussion on the practical application of the MRPS. For instance, briefly discuss its potential use in monitoring intervention processes or in adaptive training programs.

Throughout: To avoid potential confusion, briefly clarify the relationship between the terms "psychometric properties" and "initial validity evidence" in the discussion, noting that the former encompasses the latter.

  1. Conclusion (P10)

The conclusion is effective but could be slightly strengthened by adding a forward-looking statement. For example, suggest that future research should test the scale's predictive validity in longitudinal intervention studies.

Reviewer 3 Report

Comments and Suggestions for Authors

The manuscript entitled "The mindful reappraisal of pain scale: Initial validity evidence for use as a state measure following isometric exercise" aimed to evaluate the psychometric properties of the MRPS during isometric exercises. The manuscript is well structured, but it needs to be revised in the methos and the results in several points to make it more understandable:

  • I suggest adding more information to the original scale. For example, what was the validation population, and how big was that?
  • It also needs rotation to understand why this scale needs to be validated during exercise.
  • Another limitation is that the study did not include any physiological markers of exertion or metabolic stress, such as blood lactate measurement, during the isometric exercise tasks.
  • Please add the answer categories of the scales.
  • Please create a new table for the reliability measures for all the measures of this study.
  • I also suggest using their reliability measures for the MRPS. (e.g., Average Variance Extracted)
  • Since the study introduces MRPS's psychometric properties, I suggest showing age and gender differences as well.

Round 2

Reviewer 2 Report

Comments and Suggestions for Authors

The authors have addressed most of the previous concerns, and the manuscript is nearly acceptable for publication. However, the following minor revisions are still required:

  1. Please move Table 2 (which summarizes the measurement scales) to the beginning of the table section.
  2. To reduce the number of tables, please combine related tables (such as Table 4 and Table 5) into a single table, using sub-tables (a, b, etc.) for a clearer and more reader-friendly presentation.
  3. Ensure all tables follow a consistent style in terms of width, lines, and formatting.
  4. Carefully check all table content, including full explanations for any abbreviations used.
